# Economic Aspects of Zinc Oxide Fertilization in Yam (*Dioscorea alata* L.) in a Semi-Arid Region of Brazil

**Antônio Lourenço Bezerra [1], João Everthon da Silva Ribeiro [1,*], Ester dos Santos Coêlho [1], Elania Freire da Silva [1], Pablo Henrique de Almeida Oliveira [1], Gisele Lopes dos Santos [1], Antonio Gideilson Correia da Silva [1], José Travassos dos Santos Júnior [1], Ivanice da Silva Santos [1], Felipe Alves Reis [2], Lindomar Maria da Silveira [1], Aurélio Paes Barros Júnior [1] and Adriano do Nascimento Simões [2]**

[1] Department of Agronomic and Forest Sciences, Universidade Federal Rural do Semi-Árido, Mossoró 59625-900, Brazil; lourencoagronomia@gmail.com (A.L.B.); estersantos12@hotmail.com (E.d.S.C.); elania.freire23@gmail.com (E.F.d.S.); pabloalmeidaagro@gmail.com (P.H.d.A.O.); gisele1612@gmail.com (G.L.d.S.); antoniogideilson@hotmail.com (A.G.C.d.S.); travassosjunior96@gmail.com (J.T.d.S.J.); ivanice.santos@alunos.ufersa.edu.br (I.d.S.S.); lindomarmaria@ufersa.edu.br (L.M.d.S.); aurelio.barros@ufersa.edu.br (A.P.B.J.)

[2] Academic Unit of Serra Talhada, Federal Rural University of Pernambuco, Serra Talhada 55810-700, Brazil; felipe.reis@ufrpe.br (F.A.R.); adrianosimoesuast@gmail.com (A.d.N.S.)

\* Correspondence: j.everthon@hotmail.com; Tel.: +55-83-98171-6327

**Abstract:** The management and improvement of yam productivity are associated with a good supply of essential nutrients for the growth and development of the crop that has economic viability. This research aimed to evaluate the economic feasibility of foliar fertilization with Zintrac® in two yam agricultural seasons (2022/2023 and 2023/2024). Therefore, two experiments were conducted at the Rafael Fernandes Experimental Farm, Mossoró, RN, Brazil. The experimental design was in a Latin square design with five treatments of doses of Zintrac® (0, 1, 2, 3, and 4 L ha$^{-1}$) and five replications. Among the production costs of yams, labor and seed acquisition were the most significant. The highest profitability index was achieved with the dose of 1L Zintrac® ha$^{-1}$ in the first season and second season, which corresponded to 78.97 and 57.86%. For the first season, increments were observed in all treatments that received zinc doses with increases of 48.70, 31.22, 14.30, and 15.93% for 1, 2, 3, and 4 L of Zintrac® ha$^{-1}$ compared to the dose of 0 L ha$^{-1}$. On the other hand, in the second season, there was an increase only in the dose of 1 L ha$^{-1}$ of Zintrac®, which corresponded to 51.3% in the net yield (ha$^{-1}$) of the dose of 0 L ha$^{-1}$. Therefore, foliar zinc oxide fertilization was economically viable for the yam crop, obtaining higher economic indices at the dose of 1 L ha$^{-1}$. The highest cost for growing yams is using a dose of 4 L ha$^{-1}$ of Zintrac®, totaling USD 6977.59 (first season) and USD 6868.33 (second season)

**Keywords:** economic viability; tuber production; yield; production costs; foliar fertilization

## 1. Introduction

Achieving agricultural systems with high yield and productivity has been a critical factor for socioeconomic development [1]. In this context, plant mineral fertilization is the primary strategy for higher production [2]. However, the increase in the price of fertilizers requires studies that aim to indicate the efficiency and economic viability of nutrients for each crop in a specific way [2,3].

Among the crops favored by the management of applied nutrients, yam (*Dioscorea alata* L.) is an example [4]. This crop is an important food source for humans; the consumption of its tuber confers health benefits due to its nutritional benefits and herbal properties [5]. The interest of producers and the extent of cultivation has been associated with the storage potential of this species for export, which lasts on average up to 4 months under appropriate conditions without losing its viability and nutritional characteristics [6,7].

Yams are integrated among the primary root and tuber crops, mainly in Africa [8,9]. Heller et al. [10] report an increase in planted areas since yams are considered a staple food in these regions, but yields are stagnated. There are multiple aspects involved in this problem, mainly armed conflicts and extreme weather events [11].

Nutritional management to improve yam productivity is associated with a good supply of essential nutrients for crop growth and development that are economically viable since most yam farming systems are carried out with scarce resources [12,13]. Few studies have been carried out on the fertility of yams, so there is a need for research that indicates the appropriate dose with greater profitability and economic return for producers, aiming at the conscious intensification of the use of inputs [13–15].

Notably, the higher nutritional requirement of yams is associated with the macronutrient's nitrogen, phosphorus, and potassium [16], confirming Dare et al.'s (2014) [17] report that the equivalent intake of 90 kg of N, 40 kg of P, and 74 kg of K ha$^{-1}$ increased tuber production. On the other hand, research has already shown the essentiality of applying micronutrients to tubers [18,19]. Thus, the productive success of yams depends on the adequacy of agronomic practices, including nutritional management adapted to each producing region [20].

Zinc (Zn) is one of the essential nutrients for the development and growth of yams. This nutrient is a structural component of several proteins and is used as a cofactor of the main enzymes (isomerases, hydrolases, oxidoreductases, ligases, transferases, and lyases), indicating its biochemical importance [21,22]. As a structural component of antioxidant enzymes (e.g., superoxide dismutase), Zn acts directly on plant defense to eliminate free radicals under stressed conditions [23,24]. Therefore, Zn deficiency can cause metabolic disorders that compromise plant vitality [25].

However, the excess of Zn can have toxic impacts on plants' physiological processes, such as the deficiency of other essential nutrients with ionic similarity, interfering with their absorption and assimilation [26,27]. In addition, the toxicity caused by excess Zn can induce the appearance of chlorosis in leaves, reduce growth, and destabilize the photosynthetic process [28,29].

One strategy to avoid problems related to adding zinc outside the ideal range is to conduct agroeconomic studies, testing doses to indicate economic viability for the specific crop. John et al. [30] emphasize the importance of Zn for the quality and yield of tubers. The variability of yam responses to Zn fertilization indicates the need for studies to adjust the optimal level of this nutrient for the most diverse production conditions, ensuring the highest economic return [31]. For yams, more studies still need to be performed to correlate adequate doses of Zn and economic viability. Thus, this study aimed to evaluate the economic feasibility of foliar fertilization with Zn oxide in two yam growing seasons.

## 2. Materials and Methods

### 2.1. Field Conditions

The experiments were conducted in two agricultural seasons from June 2022 to March 2023 and June 2023 to March 2024 at the Rafael Fernandes Experimental Farm (5°03′31.00″ S, 37°23′47.57″ W, and 80 m altitude), belonging to the Federal Rural University of the Semi-Arid (UFERSA) in the district of Alagoinha, rural area of the municipality of Mossoró, RN.

The São Tomé yam variety was used in the experiments. This variety has an average cycle of 230 days between sowing and harvesting and does not require staking for plant development.

The experimental design was in a Latin square (5 × 5) composed of five treatment doses of Zintrac® (0, 1, 2, 3, 4 L ha$^{-1}$) and five replications. Each experimental unit had an area of 24 m$^2$, which contained 32 plants distributed in 4 rows of 5 m in length, with spacing between plants of 0.60 m × 1.20 m. The two central lines, discarding one plant at each end, were considered the useful area of the experimental unit, which contained 8.64 m$^2$.

The climate of the experiment region is characterized as BSh, tropical semi-arid hot, with an average temperature of 27.4 °C and irregular annual rainfall, with an average of

673.9 mm [32]. Meteorological data were obtained during the two crop cycles through an Automatic Weather Station installed on the experimental farm, as shown in Figure 1.

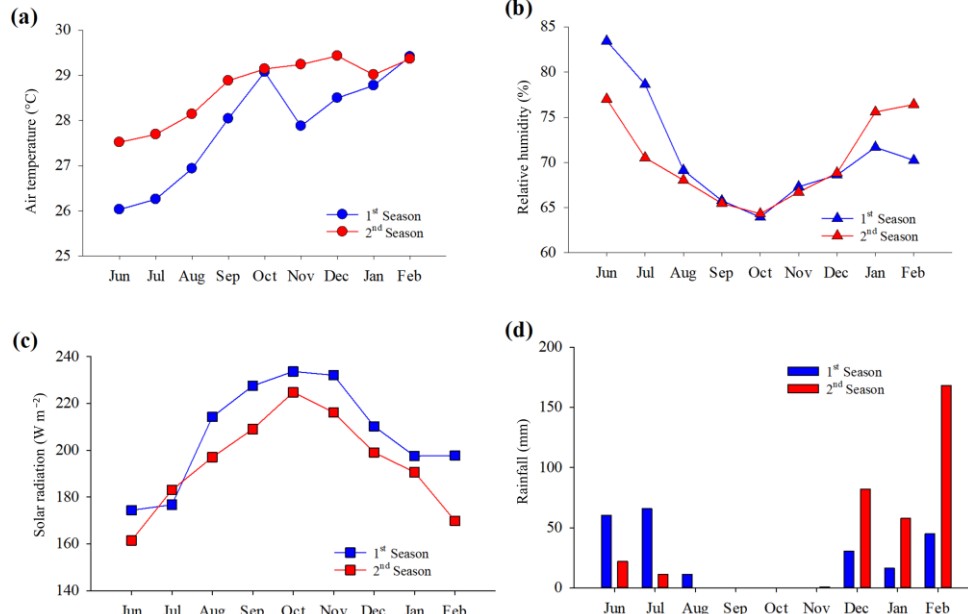

**Figure 1.** Average air temperature (**a**), relative humidity (**b**), solar radiation (**c**), and rainfall (**d**) in the two yam agricultural seasons (☐ first season: 2022/2023; ☐ second season: 2023/2024), Mossoró, RN.

The soil is classified as a Typical Dystrophic Red Ultisol [33], and the chemical and physics characteristics of the soil, determined before the installation of the experiment and according to the methodology proposed by Silva [34] and Donagema et al. [35], are presented in Table 1. The soil's physicochemical analysis was carried out at the Soil, Water, and Plant Analysis Laboratory (LASAP) of the UFERSA. The phosphorus and potassium content were higher in the first season, with increases of approximately 50% (Table 1).

**Table 1.** Chemical and physical characterization of the soil at depths of 0–0.20 m and 0.20–0.40 m in the experimental areas, referring to the two agricultural seasons (first season: 2022/2023; second season: 2023/2024).

| Depth | pH | Zn | P * | K⁺ | Na⁺ | Ca²⁺ | Mg²⁺ | Sand | Silt | Clay |
|---|---|---|---|---|---|---|---|---|---|---|
| m | | | mg dm$^{-3}$ | | | cmolc dm$^{-3}$ | | kg kg$^{-1}$ | | |
| First Season (2022/23) | | | | | | | | | | |
| 0–0.20 | 7.7 | 0.4 | 11.7 | 85.6 | 13.6 | 1.3 | 0.7 | 0.89 | 0.02 | 0.09 |
| 0.20–0.40 | 7.1 | 0.2 | 9.5 | 56.7 | 11.7 | 0.8 | 0.6 | 0.91 | 0.02 | 0.07 |
| Second Season (2023/24) | | | | | | | | | | |
| 0–0.20 | 6.6 | 0.5 | 5.6 | 40.4 | 6.3 | 0.85 | 1.09 | 0.89 | 0.02 | 0.09 |
| 0.20–0.40 | 6.6 | 0.2 | 3.7 | 35.6 | 5.3 | 0.48 | 0.68 | 0.90 | 0.02 | 0.08 |

* Element extracted with the Mehlich$^{-1}$ extractor.

Soil preparation was carried out with subsoiling, heavy harrowing to incorporate the remaining plant material, and leveling harrowing to homogenize the soil surface before the experiments were conducted.

Sowing was performed manually through sectioned tubes weighing between 100 and 150 g, distributed in holes with a depth of 5 to 8 cm. In both seasons, phytosanitary control was carried out using manual weeding and chemical control of the fungus *Curvularia eragrostidis* with foliar applications of the product based on Flutriafol (Tenaz®, Sumitomo Chemical, Maracanaú, Brazil).

Fertilization followed the recommendation of Cavalcanti et al. [36], based on soil analyses. In the first season, 80 kg ha$^{-1}$ of nitrogen (N), 70 kg of phosphorus (P), and 60 kg ha$^{-1}$ of potassium (K) were applied. In the second season, 80 kg ha$^{-1}$ of nitrogen (N), 100 kg ha$^{-1}$ of phosphorus (P), and 90 ha of potassium (K) were distributed in fertile irrigation. Urea (45% N), monoammonium phosphate (MAP) (50% $P_2O_5$), and potassium chloride (60% $K_2O$) were used as nutrient sources.

### 2.2. Characteristics of Fertilizer

Fertilizers used in the research were purchased in commercial agricultural products houses located in the city of Mossoró, RN, and contained the technical characteristics shown in Table 2.

**Table 2.** Technical data on the fertilizer used in the research, including trade name, active ingredients, density, and electrical conductivity.

| Commercial Name | Active Principle | Recommended Dose | | Density | Electrical Conductivity |
|---|---|---|---|---|---|
| | | Smaller | Bigger | (g mL$^{-1}$) | (mS cm$^{-1}$) |
| Zintrac® | Zinc oxide 40% + 1% urea | 0.25 | 2 | 1.734 | 0.24 |

### 2.3. Application of Treatments, Calculations, and Forecrops

The treatments and doses of zinc oxide at concentrations of 0, 1, 2, 3, and 4 L ha$^{-1}$ were divided into two applications the first 4 months after the installation of the experiment and the second 5 months after planting the crop. A 20 L knapsack sprayer and a spray volume equivalent to 250 L ha$^{-1}$ were used. The applications were carried out from 8:00 a.m. to 10:00 a.m.

After eight months of planting, seasons were performed manually, and commercial tubers (greater than 500 g) and seeds (less than 500 g) were collected in the two agricultural seasons. The yield of commercial tubers and seeds was determined by weighing on an analog scale, the material harvested in the useful area estimated in t ha$^{-1}$ (Table 3).

**Table 3.** Productivity of commercial yam roots fertilized with zinc oxide via foliar in two seasons (first season: 2022/2023; second season: 2023/2024).

| Zinc Doses (L ha$^{-1}$) | First Season (2022/2023) | Second Season (2023/2024) | Average of Seasons |
|---|---|---|---|
| Yield of commercial yam tubers (t ha$^{-1}$) | | | |
| 0 | 15.76 | 8.81 | 12.28 |
| 1 | 25.73 | 12.64 | 19.18 |
| 2 | 20.58 | 8.70 | 14.64 |
| 3 | 17.60 | 8.64 | 13.12 |
| 4 | 17.85 | 5.13 | 11.49 |
| Yield of yam seed (t ha$^{-1}$) | | | |
| 0 | 10.9 | 9.45 | 10.18 |
| 1 | 11.05 | 9.06 | 10.06 |
| 2 | 9.77 | 9.97 | 9.87 |
| 3 | 8.69 | 9.93 | 9.31 |
| 4 | 8.22 | 8.91 | 8.57 |

Economic indicators were evaluated to estimate the production costs of one hectare of yams at the end of each cultivation based on a methodology proposed by Conab [37]. To determine the expenses, variable costs (labor, fertilizers, and others), administrative expenses, technical assistance, rural land tax, and financial expenses were analyzed, as

well as fixed costs (depreciation and periodic maintenance of improvements/facilities) and remuneration.

Administrative and technical assistance expenses corresponding to 3 and 2% of the total cost of the crop were adopted. The fixed amount of USD 2.00 was considered as the minimum to be paid in rural land tax (RLT) in an agricultural year using Equation (1) as follows:

$$\text{RLT (USD ha}^{-1}) = \text{RLT value (USD)} \times [\text{culture cycle (days)}/365] \qquad (1)$$

The interest on the financing was attributed to the resources necessary to fund the crop, and the rate (7.49% year$^{-1}$) corresponded to the time of release or use of the capital, calculated according to Equation (2) as follows:

$$\text{Fees (USD ha}^{-1}) = \text{cost value (USD ha}^{-1}) \times [\text{culture cycle (days)}/365] \times 7.49\% \qquad (2)$$

To calculate the depreciation of the improvements and installations of the irrigation system for one hectare of yams, the use of 8400 m of low-density polyethylene drip tapes was considered, with 0.20 m spacing between emitters and a nominal diameter of 16 mm (value of the new good = USD 0.076 m$^{-1}$) with a useful life of two years. In addition to PVC pipes and fittings (value of the new good = USD 284.65), a 3.0 hp motor pump set (value of the new good: USD 590.00) with a durability of sixteen years was used. The measurement was made by Equation (3) as follows:

$$\text{Depreciation (USD ha}^{-1}) = (\text{value of the new asset (USD ha}^{-1})/\text{useful life of the asset}) \times \text{culture cycle (days)}/365 \qquad (3)$$

For periodic maintenance of the facilities and the irrigation system, a maintenance rate of 1% was adopted using Equation (4) as follows:

$$\text{Maintenance (USD ha}^{-1}) = \text{value of the new asset (USD ha}^{-1}) \times \text{culture cycle (days)}/365 \times 1\% \qquad (4)$$

Considering that the producer's investment must be remunerated as if the capital were invested in any other alternative investment, the remuneration was calculated by adopting the rate of return of 6% using Equation (5) as follows:

$$\text{Remuneration (USD ha}^{-1}) = \text{value of the new asset (USD ha}^{-1}) \times \text{culture cycle (days)}/365 \times 6\% \qquad (5)$$

Based on these data, the gross income (GI), net income (NI), rate of return (RR), and profitability index (PI) were evaluated. The rates and prices used in this study were based on information obtained through local surveys and from Embrapa's business sector.

The GI was obtained by multiplying the yield of commercial tubers and yam seeds (Y) of each treatment by the value of each product (V) paid in kilograms to the producer for USD 6.40 commercial tubers (GI = Y × V). The NI was calculated by subtracting the total production costs (PC) from inputs plus services (NI = GI − PC) from the gross income. The RR was obtained by the ratio between the gross income (GI) and the total production costs (PC) of each treatment (RR = GI/TC). The PI expressed as a percentage will be obtained by the ratio between the net income (NI) and gross income (GI).

### 2.4. Statistical Calculations

The data were submitted to analysis of variance (ANOVA), and the agricultural seasons were evaluated separately. Regression analysis and graphing were performed using the Sigmaplot 12.5 software.

## 3. Results and Discussion

### 3.1. Production Costs

The total costs for cultivating one hectare of yams, considering the maximum dose of 4 L ha$^{-1}$ of zinc oxide, were USD 6977.59 ha$^{-1}$ in the first season and USD 6868.33 ha$^{-1}$ in the second (Table 4). These values are close to the cost found in Cruz das Almas-BA in 2017, which was USD 6517.51 [38]. Variable costs that fell on top of total costs averaged 92.98%, while fixed costs accounted for 3.94%. Comparing these results with the season costs in Bonito-PE in 2018, we have values close to the variable costs that fall on the total costs, which was, on average, 97.89% [38]. Therefore, inputs and labor are among the highest costs for the implementation of the yam crop. The higher cost of the first season compared to the second is associated with the increase in fertilizer prices caused by the beginning of the war between Russia and Ukraine in 2022.

**Table 4.** Variable and fixed cost coefficients in producing one irrigated hectare of yam (*Dioscorea alata* L.), cultivated with different doses of zinc oxide, in two seasons (first season: 2022/23; second season: 2023/24).

| Discrimination | | First Season | | | Second Season | | |
|---|---|---|---|---|---|---|---|
| | Unit | Amount | Unit v. (USD) | Total (USD) | Amount | Unit v. (USD) | Total (USD) |
| I—COSTING EXPENSES | | | | | | | |
| 1. Machine rental | | | | | | | |
| Tractor with plow harrow. Subsoiler and leveling grid | h | 3 | 24 | 72.00 | 3 | 24 | 72.00 |
| 2. Labor | | | | | | | |
| Survey of the windrows | daily | 10 | 10 | 100.00 | 10 | 10 | 100.00 |
| Assembling the irrigation system | daily | 6 | 10 | 60.00 | 6 | 10 | 60.00 |
| Pickets | unit | 25 | 0.2 | 5.00 | 25 | 0.2 | 5.00 |
| Opening of the pits | daily | 2 | 10 | 20.00 | 2 | 10 | 20.00 |
| Manual planting | daily | 10 | 10 | 100.00 | 10 | 10 | 100.00 |
| Manual weeding | daily | 200 | 10 | 2000.00 | 200 | 10 | 2000.00 |
| Manual harvest | daily | 12 | 10 | 120.00 | 12 | 10 | 120.00 |
| 3. Seeds | | | | | | | |
| Acquisition of seed tubers | kg | 2900 | 0.8 | 2436.00 | 2900 | 0.8 | 2436.00 |
| 4. Crop maintenance | | | | | | | |
| Irrigation and fertigation | monthly | 5 | 30 | 150.00 | 5 | 30 | 150.00 |
| Application of pesticides (Tenaz) | daily | 8 | 10 | 80.00 | 8 | 10 | 80.00 |
| 5. Fertilizers | | | | | | | |
| Urea (46% N) | kg | 151.07 | 1.15 | 173.73 | 145.2 | 1.15 | 166.98 |
| Potassium chloride (60% K$_2$O) | kg | 104.96 | 1.50 | 157.44 | 115.5 | 1.50 | 173.25 |
| MAP (61% P$_2$O$_5$ and 12% N) | kg | 120.46 | 2.83 | 340.90 | 172.1 | 2.83 | 487.04 |
| 6. Pesticides | | | | | | | |
| Fungicide Tenaz | liter | 0.34 | 32.8 | 11.15 | 0.34 | 32.8 | 11.15 |
| 7. Other expenses | | | | | | | |
| Electrical energy for irrigation | kWh | 2214.382 | 0.07 | 155.00 | 2325 | 0.07 | 162.75 |
| Soil analysis | unit | 1 | 11.2 | 11.2 | 1 | 11.2 | 11.2 |
| Total crop costing expenses (A) | | | | 5881.82 | | | 5782.14 |

Table 4. *Cont.*

| Discrimination | First Season | | | | Second Season | | |
|---|---|---|---|---|---|---|---|
| | Unit | Amount | Unit v. (USD) | Total (USD) | Amount | Unit v. (USD) | Total (USD) |
| II—OTHER EXPENSES | | | | | | | |
| 8. Administrative expenses (3% of crop costs) | | | | 176.45 | | | 173.46 |
| 9. Technical assistance (2% of crop costs) | | | | 117.63 | | | 115.64 |
| 10. Rural land tax (USD 10.00 year$^{-1}$) | | | | 1.42 | | | 1.47 |
| TOTAL OTHER EXPENSES (B) | | | | 295.51 | | | 290.57 |
| III—FINANCIAL EXPENSES | | | | | | | |
| 11. Financing interest (7.49% year$^{-1}$) | | | | 312.61 | | | 303.75 |
| Total financial expenses (C) | | | | 312.61 | | | 311.90 |
| Variable cost (A + B + C = D) | | | | 6489.94 | | | 6384.62 |
| IV—DEPRECIATION | | | | | | | |
| 12. Depreciation of improvements/facilities | | | | 265.29 | | | 262.22 |
| Total depreciation (E) | | | | 265.29 | | | 262.22 |
| 13. Periodic maintenance of improvements/facilities (1% year$^{-1}$) | | | | 10.73 | | | 10.61 |
| Total other fixed costs (F) | | | | 10.73 | | | 10.61 |
| Fixed cost (E + F = G) | | | | 276.02 | | | 272.83 |
| Operational cost (D + G = H) | | | | 6765.97 | | | 6657.46 |
| VI—FACTOR INCOME | | | | | | | |
| 14. Expected remuneration on fixed capital (6% year$^{-1}$) | | | | 64.42 | | | 63.67 |
| Total factor income (I) | | | | 64.42 | | | 63.67 |
| Total cost (H + I = J)—0 L/ha$^{-1}$ of Zintrac® | USD | | | 6830.39 | USD | | 6721.134 |
| Total cost—1 L/ha$^{-1}$ of Zintrac® | USD | | | 6927.19 | USD | | 6817.93 |
| Total cost—2 L/ha$^{-1}$ of Zintrac® | USD | | | 6943.99 | USD | | 6834.73 |
| Total cost—3 L/ha$^{-1}$ of Zintrac® | USD | | | 6960.79 | USD | | 6851.53 |
| Total cost—4 L/ha$^{-1}$ of Zintrac® | USD | | | 6977.59 | USD | | 6868.33 |

Among the variable cost indicators, labor costs accounted for 41.24% of the total crop costs, followed by seeds (39.78%), fertilizers (10.62%), machinery rental (1.23%), and pesticides (0.2%).

Labor, therefore, represents a considerable expense in the production costs of yams since it needs high demand in activities such as cultural treatments, irrigation management, and harvesting. Specifically, in properties that do not have machinery and pesticides for mechanized cultivation, this increases the number of contractors and, consequently, the costs of labor.

Seeds have the second largest participation in variable costs, with a high market value ranging from 0.40 to 0.80 USD kg$^{-1}$. This price variation depends on market availability. Usually, producers choose to use their own seeds, making the purchase only at the beginning of the economic activity, which favors lower costs.

Fertilizers were third among the variable costs requiring attention because, in most crops, inputs represent a high cost, as reported by Sarker and Alam [39], who emphasize inputs as the main alarming factor of high costs, especially among small farmers.

The contribution of foliar zinc oxide fertilization to production costs was in the order of 0, 1.42, 1.66, 1.91, and 2.15% for doses of 0, 1, 2, 3, and 4 L ha$^{-1}$ in the first season. At the same time, in the second season, the costs corresponded to 0, 1.44, 1.69, 1.94, and 2.19% for

the same doses, with an increase between the values in this second cycle; this increase is justified by the decrease in fertilizer prices throughout the second season. The contribution values of the fertilizer to the costs of the two agricultural seasons are low, indicating the feasibility of its use by the producer.

The use of pesticides was low between seasons, contributing to a lower participation in production costs. However, this is a factor dependent on the manifestation of pests and diseases in crops, and there may be a significant change in the profile of costs attributed to the use of pesticides. In addition, the factors that interfere in forming agricultural prices are different and can even include uncontrollable conditions, such as edaphoclimatic conditions.

In this sense, the higher yield per hectare will not always characterize higher profit, especially considering that it is possible to lower costs using a smaller amount of inputs in the crop [40].

*3.2. Gross Income*

The highest gross income rates were observed in the first season, with an average reduction of approximately 55% in the first season in relation to the second season (Figure 2). This fact is related to the high yield found when compared with the second season (Table 3), which possibly occurred due to the low levels of nutrients found in the soil analysis of the area in the second season (Table 1).

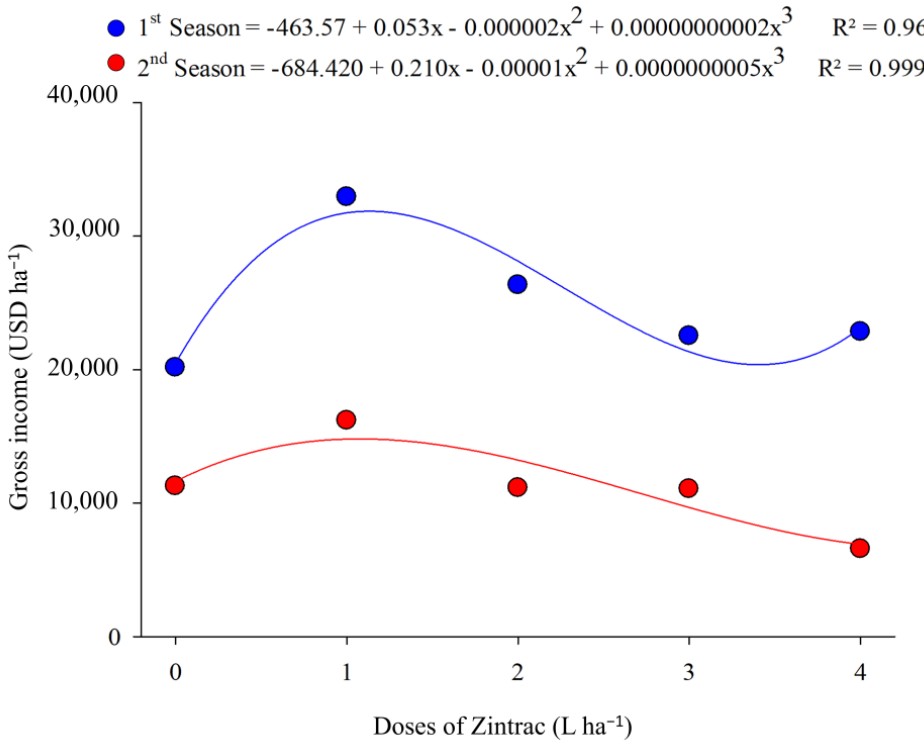

**Figure 2.** Gross income of two agricultural seasons (● first season: 2022/2023; ● second season: 2023/2024) of the yam cultivation under doses of Zintrac®, Mossoró, RN.

The values found in this research indicate that depending on the area and nutrient levels of the soil, the gross income can drop drastically because within the same soil, mechanical tillage and nitrogen fertilizer management over time can alter the behavior of its biological components and, consequently, the use of nutrients by cultivated plants [41]. In addition, gross income may change with each season, as observed in this study, as planting variables obey several situations that must be periodically observed to improve the process [38]. The dose of 1 L ha$^{-1}$ provided the highest yield in both crops, indicating that it was the best treatment in the yam crop.

### 3.3. Net Income

The highest net yields were evidenced in the dose of 1 L ha$^{-1}$ of Zintrac$^{®}$ in both agricultural seasons with increments of 48.70 and 51.34%, respectively, in the first and second agricultural seasons compared to their respective treatments that did not receive zinc doses (Figure 3). For the first season, increments were observed in all treatments that received zinc doses with increases of 48.70, 31.22, 14.30, and 15.93% for 1, 2, 3, and 4 L Zintrac$^{®}$ ha$^{-1}$ at a dose of 0 L ha$^{-1}$. On the other hand, the second season showed an increase only up to a dose of 1 L Zintrac$^{®}$ ha$^{-1}$, showing a difference of USD 4.80 in net income (ha$^{-1}$) (Figure 3). An explanation for these results for a dose of 1 L ha$^{-1}$ may be related to the absorption of other nutrients since excessive doses of Zn interfere with the absorption of essential elements, resulting in heavy metal toxicity [24].

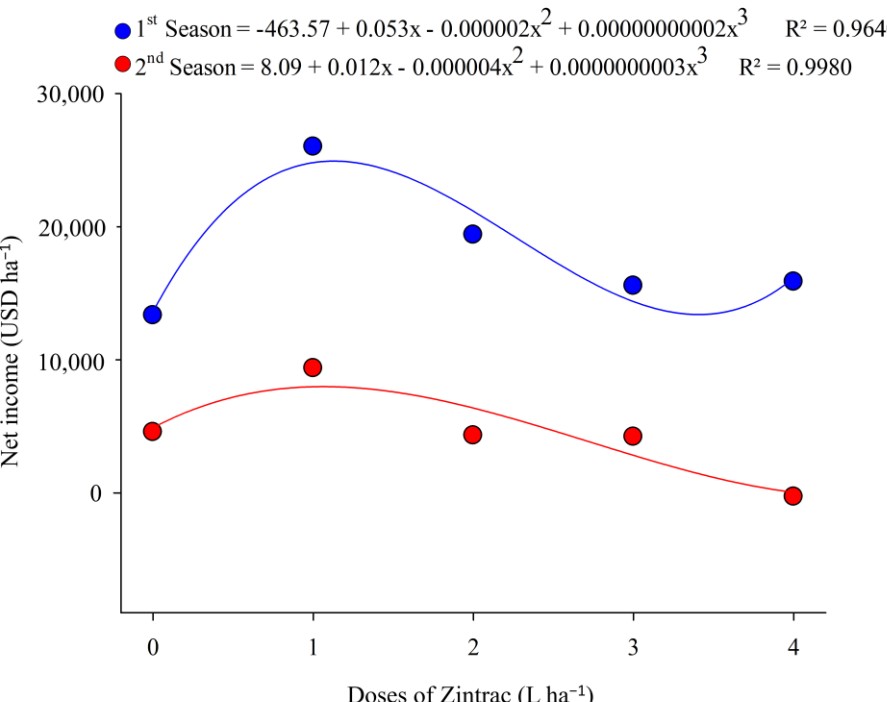

**Figure 3.** Net income from two agricultural seasons (● first season: 2022/2023; ● second season: 2023/2024) of the yam crop under doses of Zintrac$^{®}$, Mossoró, RN.

Findings in the literature emphasize that fortification with Zn is an easy-to-use and very profitable technology; in addition, it is an important alternative, especially for tuber crops, because they are staple foods and have a good amount of nutritional content, thus prioritizing vulnerable populations [30]. Banerjee et al. (2016) [42] obtained an increase in net income in their research with *Solanum tuberosum* as Zn doses increased, mainly up to a dose of 4.5 kg ha$^{-1}$ of Zn. Das and Chakraborty [43] tested Zn doses in *Solanum tuberosum* and showed a net income of USD 981.91 (ha$^{-1}$) at a 6.0 ha$^{-1}$ dose of Zn. Furthermore, the authors mention that Zn is highly affected by soil factors, pH, clay content, and modifications in anatomical structures in the conductive tissue, affecting the root/sprout ratio.

### 3.4. Rate of Return

In the present study, the return rate results showed a better fit using a third-degree polynomial ($p \leq 0.05$). The absence of Zintrac$^{®}$ doses (0 L ha$^{-1}$) shows a significant percentage reduction in both agricultural seasons (Figure 4). The comparative analysis of the rates of return between the first season and second season, with the application of 1 L ha$^{-1}$ of Zintrac$^{®}$, revealed a considerable decrease in profitability. In the first season, the rate of return reached 2.95%, while in the second season, this percentage was reduced to

1.68%. This difference represents a significant decrease of approximately 43% in profitability between the two seasons, indicating a marked influence of the agricultural cycle on the returns obtained by applying this input. There was a decreasing trend in the rate of return as the doses of Zintrac® were increased in the first season. However, between the doses 3 L ha$^{-1}$ (3.23%) and 4 L ha$^{-1}$ (3.27%), a slight increase in the return rate of approximately 1.24% was observed. On the other hand, in the second season, the decrease in the return rate was gradual as the doses increased, and the dose of 4 L ha$^{-1}$ registered a lower value than the control (0 L ha$^{-1}$).

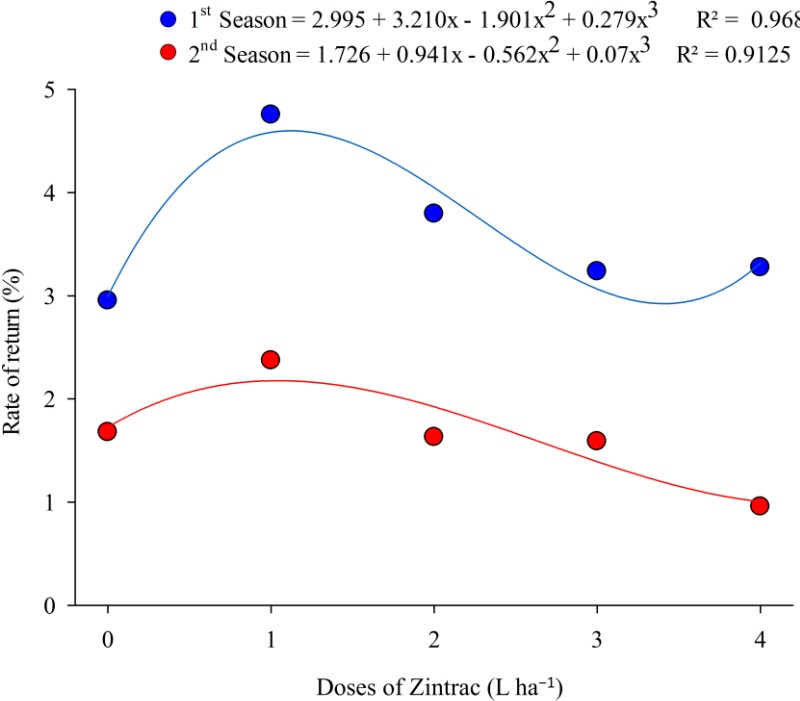

**Figure 4.** Rate of return of two agricultural seasons (● first season: 2022/2023; ● second season: 2023/2024) of the yam cultivation under doses of Zintrac®, Mossoró, RN.

The analysis of the results of the present study revealed a complex relationship between the application of Zintrac® doses and the rate of return in the agricultural seasons studied. The absence of application of Zintrac® doses implied a significant reduction in the percentages of return in both agricultural seasons, indicating that the lack of application of inputs can negatively affect the profitability of the crop [44]. The variation in the rate of return between seasons with the application of Zintrac® doses can be attributed to several factors, including climatic variations, soil characteristics, and agronomic practices specific to each growing period [45]. The decrease in the rate of return with the increase in the dose in the first season indicates a possible saturation of the positive effect of Zintrac®. In addition, the increase in the rate of return at the dose of 4 L ha$^{-1}$ in the second season can be explained by the nonlinear response of the crop to nutrient availability, where moderate doses can lead to an increase in yield. In contrast, excessive doses can decrease economic return [46].

*3.5. Profitability Index*

Regarding the profitability index (Figure 5), it was observed that the maximum value (78.97%) was reached with a dose of 1 L ha$^{-1}$ of zinc in the first season, an increase corresponding to 16.25% when compared to a dose of 0 L ha$^{-1}$ of zinc. In the second season, the highest index (57.86%) was also obtained with a dose of 1 L ha$^{-1}$ of zinc; however, it was 26.73% less than in the previous season but had an increase of 30.17% for a dose of 0 L ha$^{-1}$ of zinc. These differences between the agricultural seasons were possibly due to the edaphoclimatic variations between them, which are shown in Figure 1 and Table 2 and

mainly consider higher temperatures and rainfall, as well as lower soil pH and lower levels of nutrients, such as P, K, and Ca, recorded in the second season. These variations affect crop management practices, implying higher costs and, consequently, affecting profitability.

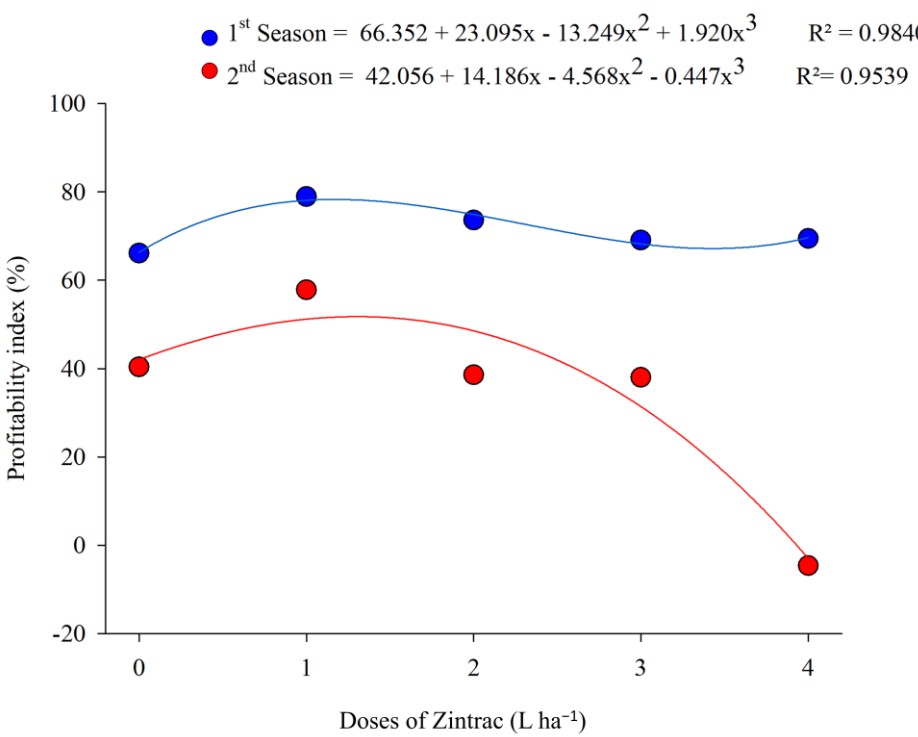

**Figure 5.** Profitability index (%) of two agricultural seasons (● first season: 2022/2023; ● second season: 2023/2024) of yam cultivation under doses of Zintrac®, Mossoró, RN.

Variations in edaphoclimatic parameters between seasons were pointed out in a study developed in the same region as influencing the agronomic characteristics of different crops in the field [47], corroborating what was mentioned in this work. According to Santos et al. [48], a profitability rate of 54% for the production of yams indicates that the activity is attractive for investments by rural producers. In the case of this study, the index was higher in both seasons, indicating good economic prospects for the cultivation of yams. In a study conducted by Akintunde et al. [49] in Nigeria, the authors indicated that the production and marketing of yams are highly profitable, thus supporting the findings of this work. According to Udemezue and Nnabuife [50], among the challenges of yam producers that can affect the economy are inadequate extension services, pest and disease attacks, adverse weather conditions, high labor costs, high cost of yams, and the use of unimproved seeds.

## 4. Conclusions

A dose of Zintrac® of 1 L ha$^{-1}$ provided higher profitability rates in the two agricultural seasons, reaching values corresponding to 78.97% and 57.86% in the first and second agricultural seasons. The net income increased by about a dose of 0 L ha$^{-1}$ in the order of 94.92, 45.38, 16.67, and 18.94% for doses 1, 2, 3, and 4 L ha$^{-1}$ of Zintrac®. On the other hand, the second season showed an increase only in a dose of 1 L ha$^{-1}$. Using foliar zinc oxide fertilization was economically viable for the yam crop, obtaining higher economic indices at a dose of 1 L ha$^{-1}$. The highest cost for growing yams is using a dose of 4 L ha$^{-1}$ of Zintrac®, totaling USD 6977.59 (first season) and USD 6868.33 (second season). Given this, other future studies are needed to fill and elucidate gaps regarding zinc fertilization in yam cultivation.

**Author Contributions:** Conceptualization, A.L.B. and A.d.N.S.; methodology, A.L.B., A.d.N.S. and J.E.d.S.R.; validation, A.L.B., P.H.d.A.O., A.G.C.d.S. and J.T.d.S.J.; formal analysis, A.L.B., E.d.S.C.,

E.F.d.S. and G.L.d.S.; investigation, A.L.B., P.H.d.A.O., A.G.C.d.S., J.T.d.S.J., E.d.S.C., E.F.d.S. and G.L.d.S.; resources, A.L.B., A.d.N.S., L.M.d.S. and A.P.B.J.; data curation, A.L.B. and J.E.d.S.R.; writing—original draft preparation, A.L.B., P.H.d.A.O., E.d.S.C., E.F.d.S., G.L.d.S. and J.E.d.S.R.; writing—review and editing, J.E.d.S.R., A.d.N.S., I.d.S.S. and F.A.R.; visualization, J.E.d.S.R. and A.d.N.S.; supervision, A.d.N.S., L.M.d.S. and A.P.B.J.; project administration, A.d.N.S., L.M.d.S. and A.P.B.J.; funding acquisition, A.d.N.S. and A.P.B.J. All authors have read and agreed to the published version of the manuscript.

**Funding:** The research was supported by CAPES—Coordination for the Improvement of Higher Education Personnel (Proc. 88881-159183/2017-01); FACEPE—Foundation for the Support of Science and Technology of the State of Pernambuco (PQ-0795-5.01/16); UFRPE Federal Rural University of Pernambuco (PRPPG 015/2018), and CNPq—National Council for Scientific and Technological Development (423100/2018-1).

**Data Availability Statement:** Data are contained within the article.

**Acknowledgments:** We thank the Universidade Federal Rural do Semi-Árido for logistical support during the research.

**Conflicts of Interest:** The authors declare no conflicts of interest.

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
