# Peer review of "Economic Aspects of Zinc Oxide Fertilization in Yam (Dioscorea alata L.) in a Semi-Arid Region of Brazil"

_horticulturae, doi:10.3390/horticulturae10050489_

Round 1

Reviewer 1 Report

Comments and Suggestions for Authors

The manuscript titled "Economic evaluation of zinc oxide fertilization in yam (Dioscorea alata L.) in a semiarid region of Brazil" contains results on the profitability of cultivation and fertilization of Zn yam. Economic calculations are an important aspect in plant production. However, the manuscript requires improvement. I included detailed comments in the original text (pdf).

My general comments:

line 21. This was the dose of ZnO or Zintrac® foliar fertilizer ?

Add to keywords: foliar fertilization

Correct the cited literature in accordance with the journal's requirements

Improve English throughout, e.g.

Add new subsections to chapter 2

What variety was grown ?

Briefly describe the weather conditions in the studied seasons and soil conditions

Do you have Zn content in your soil?

What was the forecrop?

In the Material and Methods section, write what pesticides you used?

line 119. What is the recommended dose of Zintrac®?

line 132 no results for yam seed

How do you explain such a large increase in yields after using 1 liter of the product?

Explain what the stars (asterix) in table 4 mean

Write in the conclusion whether this experiment should be continued and why?

Correct the literature list

Write Latin names in italics

After making corrections, I recommend publishing the manuscript in the journal Horticulturae

Author Response

We appreciate the suggestions and contributions to improve the manuscript. All changes and suggestions within the file have been accepted. Below is the response to this reviewer:

Reviewer's comment: line 21. This was the dose of ZnO or Zintrac® foliar fertilizer ?

Reply: The correct is Zitrac®. This information has been corrected within the text and in all figures and tables.

Reviewer's comment: Add to keywords: foliar fertilization.

Reply: Done.

Reviewer's comment: Correct the cited literature in accordance with the journal's requirements

Reply: Done.

Reviewer's comment: Improve English throughout, e.g.

Reply: English has been improved. As the reviewer suggested, we are willing to revise the English once again in the final version of the manuscript.

Reviewer's comment: Add new subsections to chapter 2

Reply: Done.

Reviewer's comment: What variety was grown ?

Reply: The variety used was mentioned. Some characteristics of this variety were added in the second paragraph of M&M.

Reviewer's comment: Briefly describe the weather conditions in the studied seasons and soil conditions

Reply: Done. This information has been added.

Reviewer's comment: Do you have Zn content in your soil?

Reply: Done. The Zn content in the soil was added to the Table.

Reviewer's comment: What was the forecrop?

Reply: This information was added in the second paragraph of subtopic 3.2.

Reviewer's comment: In the Material and Methods section, write what pesticides you used?

Reply: Done. Information about the pesticide used has been added.

Reviewer's comment: line 119. What is the recommended dose of Zintrac®?

Reply: Done. This information was added to the Table.

Reviewer's comment: line 132 no results for yam seed

Reply: Yam seed results have been added to the Table.

Reviewer's comment: After applying 1 liter of fertilizer, the yield increase was 9,97 t ha?

Reply: Yes, that was the result found.

Reviewer's comment: How do you explain such a large increase in yields after using 1 liter of the product?

Reply: The concentration of 1 L ha-1 provided the best values because it provides the crop's amount needed for best development. On the other hand, the highest doses could have been more efficient. Studies on yam cultivation still need to be done using this zinc oxide-based product.

Reviewer's comment: Explain what the stars (asterix) in table 4 mean

Reply: Asterisks have been removed from the table. It could have been a typing error.

Reviewer's comment: Write in the conclusion whether this experiment should be continued and why?

Reply: Done. This information was added in the Conclusion.

Reviewer's comment: Correct the literature list

Reply: Corrected. Done.

Reviewer's comment: Write Latin names in italics

Reply: Corrected. Done.

Reviewer 2 Report

Comments and Suggestions for Authors

Dear authors,

This manuscript is titled Economic evaluation of zinc oxide fertilization in yam (Di- oscorea alata L.) in a semiarid region of Brazil.”

Such studies have been done already and are not novel. Many kinds of literature have mentioned this issue and well-known knowledge exists compared to the data from this study. Mention your significance with novelty.

-Not well structured.

-Why only zinc oxide fertilization? What about other nutrients?

 “Major Revision”

Some comments for your manuscript

-Check your spelling and grammar mistakes.

-Follow journal guidelines for the format of the manuscript.

Title:

Better to improve the title as the abstract shows no economic value.

Abstract

- According to the title Zinc oxide fertilization has economic importance. But where are values? costs? Should be in $.

-Seems it’s about yield

-Clear your methodology. Duration? Experiment details if any treatments?

- Be specific with your study.

-Make it more clear and rewrite your abstract.

Introduction

-Try to make a story instead of short paragraphs about your study. Make it a relationship as well with its economic significance.

-Mention your objectives.

-Add more related recent references.

Materials and methods

-What is the baseline for comparison?

-Please provide references for each method.

- Basic properties of all substrates. Duration?

Results and discussion

-Figure details in their captions?

-Improve all your figures with significant analysis.

-Adding more references in the discussion especially related to cost comparison will be better in your study.

-Try to make a relative analysis for yield and its costs.

Conclusions

-Too simple. Please elaborate more on increasing yield and its costs.

References

1. Fernie, A.R.; Yan, J. De novo domestication: an alternative route toward new crops for the future. Molecular plant 2019, 12, 615- 347

9. Canton, H. Food and agriculture organization of the United Nations—FAO. In The Europa directory of international organizations 364

2021; Routledge: 2021; pp. 297-305. Be consistent.

Check two references' style. Same? Then others?

-Follow journal guidelines.

Author Response

We appreciate the suggestions and contributions to improve the manuscript. Below is the response to this reviewer:

Reviewer's comment: Why only zinc oxide fertilization? What about other nutrients?

Reply: Zinc is one of the main nutrients essential for the development and growth of yam. This nutrient is a structural component of several proteins and is used as a cofactor in the main enzymes, indicating its biochemical importance. In the literature, there are still no studies with an economic evaluation of the foliar application of Zinc in yam plants.

Reviewer's comment: Better to improve the title as the abstract shows no economic value.

Reply: The title has been modified. The abstract was modified and values ​​were added.

Reviewer's comment: According to the title Zinc oxide fertilization has economic importance. But where are values? costs? Should be in $.

Reply: Done. New information with costs and values ​​has been added.

Reviewer's comment: Clear your methodology. Duration? Experiment details if any treatments?

Reply: The duration of the experiment was added in the first paragraph of topic 2.1 “Field conditions”. Information about treatments is described in topic 2.3 “Application of treatments, calculations and forecrop”.

Reviewer's comment: Be specific with your study.

Reply: New information was added to improve the manuscript.

Reviewer's comment: Make it more clear and rewrite your abstract.

Reply: This section has been modified according to reviewers' suggestions.

Reviewer's comment: Try to make a story instead of short paragraphs about your study. Make it a relationship as well with its economic significance.

Reply: New information was added to improve the manuscript.

Reviewer's comment: Mention your objectives.

Reply: The objective of the work is described at the end of the Introduction.

Reviewer's comment: Add more related recent references.

Reply: New references have been added. Most of the last five years.

Reviewer's comment: Please provide references for each method.

Reply: All calculations were carried out according to a single methodology. The reference was mentioned in the Material and Methods section.

Reviewer's comment: Basic properties of all substrates. Duration?

Reply: Duration has been added.

Reviewer's comment: Figure details in their captions?

Reply: Done.

Reviewer's comment: Improve all your figures with significant analysis.

Reply: The numbers were recalculated, corrected, and inserted into the text.

Reviewer's comment: Your study would benefit from adding more references in the discussion, especially related to cost comparison.

Reply: New information has been added. As this work is unique, we sought to compare it as much as possible with other studies.

Reviewer's comment: Try to make a relative analysis for yield and its costs.

Reply: We appreciate the suggestion, but it will not be possible to carry out this analysis. The yield was shown in this work only as a characterization and justification for the total costs.

Reviewer's comment: Too simple. Please elaborate more on increasing yield and its costs.

Reply: Information about yield and costs has been added. As another reviewer suggested, we also emphasize the importance of carrying out new studies.

Reviewer 3 Report

Comments and Suggestions for Authors

The authors examined the economic prospects of yam roots (Dioscorea alata L) that were foliar fertilized with Zinc oxide for two seasons- 2022/2023 & 2023/2024. The experiments were performed in a dose-dependent manner. The authors reported that 1L per ha of ZnO dose produced optimum yields in both seasons. The work is interesting but needs major revisions and improvement in the grammar. There are some instances where the language is unclear.

Major comments

1.     Introduction: Pg 2 Line 47. Yam yield stagnation as alluded to in West Africa is not mainly due to a lack of compliance with the fertility requirements of the crop or inadequate fertilizer rate. Recently, I would argue that insecurity, armed conflicts and extreme weather events are bigger problems for West African farmers than compliance with fertility requirements (For reference, see: https://doi.org/10.1016/j.jaridenv.2020.104398

2.     M&M: The authors stated that commercial yam roots were studied. There is a need to know what variety was studied.

3.     I also thought about the possibility of using US Dollars instead of Brazilian Real for a better understanding of the economic parameters.

4.     Since foliar fertilisation was done, I expected physiological and maybe biochemical data to support the yield data presented. For reference, see:  https://doi.org/10.56612/ijaeb.v2i2.26. In this paper, three different wheat varieties were grown for two seasons in Pakistan and physiological and biochemical data were estimated to match yield attributes

Minor comments

Table 3 title is unclear

Pg 4. Line 119 Characteristics not ‘caracteristics’

Comments on the Quality of English Language

Moderate editing of the English language is required

Author Response

We appreciate the suggestions and contributions to improve the manuscript. Below is the response to this reviewer:

Reviewer's comment: Introduction: Pg 2 Line 47. Yam yield stagnation as alluded to in West Africa is not mainly due to a lack of compliance with the fertility requirements of the crop or inadequate fertilizer rate. Recently, I would argue that insecurity, armed conflicts and extreme weather events are bigger problems for West African farmers than compliance with fertility requirements (For reference, see: https://doi.org/10.1016/j.jaridenv.2020.104398

Reply: Done. The information has been corrected. Suggested reference has been added.

Reviewer's comment: M&M: The authors stated that commercial yam roots were studied. There is a need to know what variety was studied.

Reply: The variety used was mentioned. Some characteristics of this variety were added in the second paragraph of M&M.

Reviewer's comment: I also thought about the possibility of using US Dollars instead of Brazilian Real for a better understanding of the economic parameters.

Reply: This has been done. All changes were made to the text, tables, and figures.

Reviewer's comment: Since foliar fertilisation was done, I expected physiological and maybe biochemical data to support the yield data presented. For reference, see:  https://doi.org/10.56612/ijaeb.v2i2.26. In this paper, three different wheat varieties were grown for two seasons in Pakistan and physiological and biochemical data were estimated to match yield attributes

Reply: We found the idea of ​​adding physiological and biochemical data interesting. The recommended article is excellent. Unfortunately, we are unable to add this data to the manuscript. We had some problems with these analyses, and it was impossible to insert them.

Reviewer's comment: Table 3 title is unclear

Reply: Information on seed productivity has been added. This was missing from the table.

Reviewer's comment: Pg 4. Line 119 Characteristics not ‘caracteristics’

Reply: Done.

Round 2

Reviewer 1 Report

Comments and Suggestions for Authors

The manuscript titled "Economic evaluation of zinc oxide fertilization in yam (Dioscorea alata L.) in a semiarid region of Brazil " has been well revised. All my comments have been corrected. Recommends publishing the manuscript in the journal Horticulturae. Thank you for your cooperation.

Reviewer 2 Report

Comments and Suggestions for Authors

The authors improved their manuscript and followed the comments.

Reviewer 3 Report

Comments and Suggestions for Authors

Well done